# Learning with Language-Guided State Abstractions

**Andi Peng**
MIT

**Ilia Sucholutsky**[*]
Princeton

**Belinda Z. Li**[*]
MIT

**Theodore R. Sumers**
Princeton

**Thomas L. Griffiths**
Princeton

**Jacob Andreas**
MIT

**Julie A. Shah**
MIT

## Abstract

We describe a framework for using natural language to design state abstractions for imitation learning. Generalizable policy learning in high-dimensional observation spaces is facilitated by well-designed state representations, which can surface important features of an environment and hide irrelevant ones. These state representations are typically manually specified, or derived from other labor-intensive labeling procedures. Our method, LGA (*language-guided abstraction*), uses a combination of natural language supervision and background knowledge from language models (LMs) to automatically build state representations tailored to unseen tasks. In LGA, a user first provides a (possibly incomplete) description of a target task in natural language; next, a pre-trained LM translates this task description into a state abstraction function that masks out irrelevant features; finally, an imitation policy is trained using a small number of demonstrations and LGA-generated abstract states. Experiments on simulated robotic tasks show that LGA yields state abstractions similar to those designed by humans, but in a fraction of the time, and that these abstractions improve generalization and robustness in the presence of spurious correlations and ambiguous specifications. We illustrate the utility of the learned abstractions on mobile manipulation tasks with a Spot robot.

## 1 Introduction

In unstructured environments with many objects, distractors, and possible goals, learning generalizable policies from scratch using a small number of demonstrations is challenging (Bobu et al., 2024; Correia & Alexandre, 2023). Consider the demonstration in Fig. 1A, which shows a Spot robot executing a specific maneuver to perform a desired task. Which task is demonstrated here—grabbing an object, grabbing an orange, or giving an orange to a user? Here, the observations do not provide enough evidence to meaningfully disentangle which features are relevant for downstream learning.

In humans, *abstraction* is essential for generalizable learning (McCarthy et al., 2021b). When learning (and planning), humans reason over simplified representations of environment states that hide details and distinctions not needed for action prediction (Ho et al., 2022). Useful abstractions are task-dependent, and a growing body of evidence supports the conclusion that humans flexibly and dynamically construct such representations to learn new tasks (Ho et al., 2023; Huey et al., 2023). Importantly, this process of abstraction does not begin with a blank slate—instead, experience, common-sense knowledge, and direct instruction provide rich sources of prior knowledge about which features matter for which tasks (Fan et al., 2020). In Fig. 1A, learning that the demonstrated skill involves a *fruit* combined with prior knowledge about which objects are fruits makes clear that the object's identity (*orange*) is likely important. Meanwhile, the demonstration provides complementary information (about the desired movement speed, goal placement, etc.) that is hard to specify in language and not encapsulated by the utterance *bring me a fruit* alone. In other words, *actions* also contain valuable information that is necessary to complement the abstraction.

---

[*]Equal contribution.

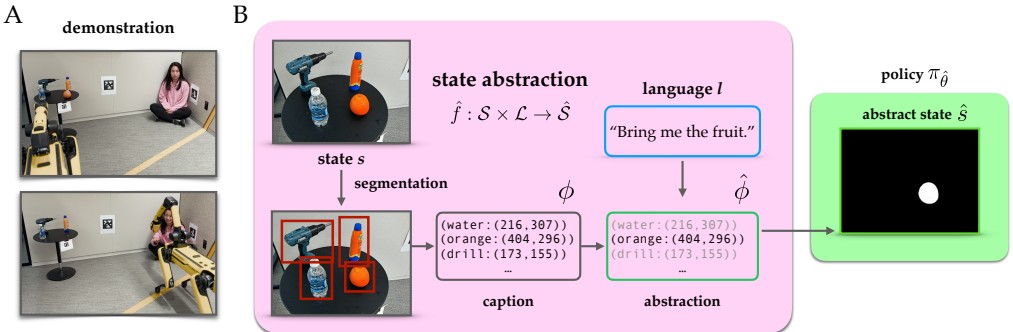

Figure 1: **A**: Example demonstration in our environment, showing Spot picking up an orange and bringing it to the user. **B**: Our approach, *Language Guided Abstraction* (LGA), creates a state abstraction with task-relevant features identified by an LM. The policy is learned directly over this abstracted state.

What would it take to build autonomous agents that can leverage both demonstrations and background knowledge to reason about tasks and representations? State abstraction has been a major topic of research from the very earliest days of research on sequential decision-making, with significant research devoted to both unsupervised representation learning (Coates & Ng, 2012; Bobu et al., 2023; Higgins et al., 2017; Lee et al., 2021) and human-aided design (Abel et al., 2018; Cakmak & Thomaz, 2012; Bobu et al., 2021; Abel et al., 2016). However, there are currently few tools for autonomously incorporating human prior knowledge for constructing state abstractions in unseen, unstructured environments.

In this paper, we propose to use *natural language* as a source of information for constructing state abstractions. Our approach, **Language-Guided Abstraction (LGA)** (Fig. 1B), begins by querying humans for high-level task descriptions, then uses a pre-trained language model (LM) to translate these descriptions into task-relevant state abstractions. Importantly, LGA requires only natural language annotations for state features. Unlike most recent work applying LMs to sequential decision-making tasks, it does not depend on pre-trained skills (Ahn et al., 2022; Huang et al., 2022b), environment interaction (Du et al., 2023), large multitask datasets (Karamcheti et al., 2023; Shridhar et al., 2022), or even the ability to describe behavior in language (Kwon et al., 2023). It complements traditional supervised learning methods like behavior cloning (BC), without relying on additional assumptions about the data labeling process. Experiments comparing LGA to BC (and stronger variants of BC) show that LGA-generated abstractions improve sample efficiency and distributional robustness in both single- and multi-task settings. They match the performance of human-designed state abstractions while requiring a fraction of the human effort.

In summary, we **(1)** introduce LGA, a method for using text descriptions and language models to build state abstractions for skill learning; **(2)** show that LGA produces state abstractions similar (and similarly effective) to those manually designed by human annotators with significantly less time; **(3)** show that LGA-constructed state abstractions enable imitation learning that is more robust to the presence of observational covariate shift and ambiguous linguistic utterances; and **(4)** demonstrate LGA's utility in real world mobile manipulation tasks with a Spot robot.

## 2 RELATED WORK

**Language-Aided Reward Design.** Large language models, trained on large amounts of text data, contain commonsense information about object properties, functions, and their salience/relevance to various tasks. Several works (Goyal et al., 2019; Kwon et al., 2023; Sumers et al., 2021; Carta et al., 2022) leverage this to shape or learn reward models by training a policy to complete intermediate or higher-level tasks and asking the LM to annotate resulting behavior. Language feedback can then be incorporated as a way to *guide* the policy during training (Mu et al., 2022; Du et al., 2023). However, just as with any reinforcement learning method, these works assume continuous environment access during the training process as the policy gradually improves with the updated reward (Shinn et al., 2023). In contrast, we instead leverage LMs to produce state abstractions for learning skills via *imitation learning*, which does not require continuous environment interaction or access to rewards.

**Language-Aided Planning.** In parallel, a great deal of work has leveraged LMs to output plans directly, i.e. generate primitives or high-level action sequences (Sharma et al., 2022; Ahn et al., 2022; Huang et al., 2022a;b).= These approaches use priors embedded in LMs to produce better *instruction following* models, or in other words, better compose pre-existing skills to generate more complex behavior (Zeng et al., 2023; Li et al., 2023; Ahn et al., 2022; Wang et al., 2023). Such methods can be thought of as performing *action abstraction* and require access to a library of pre-trained (often text annotated) skills. In contrast, we use LMs to perform *state abstraction* for learning better skills *from scratch*, and can therefore also sit upstream of any instruction following method.

**State Abstractions in Learning.** There is substantial evidence to suggest much of the flexibility of human learning and planning can be attributed to information filtering of task-relevant features (McCarthy et al., 2021a; Huey et al., 2023; Ho et al., 2022; Chater & Vitányi, 2003). This suggests flexibly creating task-conditioned abstractions is important for generalizable downstream learning, particularly in low-data regimes (Mu et al., 2022; Bobu et al., 2024). While state abstraction has been explored in sequential decision-making (Thomas & Barto, 2011), existing methods often assume pre-defined hierarchies over the state-action space of the given environment or hand defined primitives (Abel et al., 2016; Diuk et al., 2008). In this work, we explore a method to *autonomously* construct state abstractions from RGB observations with only a text specification of the task.

Perhaps the closest comparison to our work is Misra et al. (2018), which conditions on language and raw visual observations to create a *binary goal mask*, specifying goal location within an observation. However, our approach is much more flexible (our approach can learn any demonstrated behavior, not just goal navigation) and more personalizable (our approach allow humans to interact with the system, enabling them to refine the representation based on individual preference).

## 3   PROBLEM STATEMENT

**Preliminaries.** We formulate our tasks as Markov Decision Processes (MDPs) (Puterman, 2014) defined by tuples $\langle \mathcal{S}, \mathcal{A}, \mathcal{T}, \mathcal{R} \rangle$ where $\mathcal{S}$ is the state space, $\mathcal{A}$ the action space, $\mathcal{T} : \mathcal{S} \times \mathcal{A} \times \mathcal{S} \to [0, 1]$ the transition probability distribution, and $\mathcal{R} : \mathcal{S} \times \mathcal{A} \to \mathbb{R}$ the reward function. A policy is denoted as $\pi_\psi : \mathcal{S} \to \mathcal{A}$.

In **behavioral cloning (BC)**, we assume access to a set of expert demonstrations $D_{\text{train}} = \{\tau^i\}_{i=0}^n = \{(s_0^i, a_0^i, s_1^i, a_1^i, ... s_t^i, a_t^i)\}_{i=0}^n$ from which we "clone" a policy $\pi_\psi$ (Pomerleau, 1988) by minimizing:

$$\mathcal{L}_{\text{BC}} = \mathbb{E}_{(s_t^i, a_t^i) \sim D_{\text{train}}}[\|\pi_\psi(s_t^i) - a_t^i\|_2^2] . \tag{1}$$

In **goal-conditioned behavioral cloning (GCBC)**, policies additionally condition on goals $\ell$ (Co-Reyes et al., 2018). Motivated by the idea that natural language is a flexible, intuitive interface for humans to communicate, we specify the goal through language instruction $\ell \in \mathcal{L}$, resulting in:

$$\mathcal{L}_{\text{GCBC}} = \mathbb{E}_{(s_t^i, a_t^i, \ell^i) \sim D_{\text{train}}}[\|\pi_\psi(s_t^i, \ell^i) - a_t^i\|_2^2] . \tag{2}$$

Above we have highlighted differences from BC in red.

A (GC)BC policy $\pi(s_t^i, \ell^i)$ must generalize to novel commands $\ell^i$ and contextualize them against potentially novel states $s_t^i$. Due to difficulties with sampling representative data and expense of collecting a large number of expert demonstrations, systematic biases may be present in both the training demonstrations $(s_0^i, a_0^i, \cdots)$ for a goal, or the commands $\ell^i$ describing the goal. This can result in brittle policies which fail to generalize (Ross et al., 2011).

We study two sources of covariate shift: (1) out-of-distribution observations $s^i$, or (2) out-of-distribution utterances $\ell^i$. Training a policy $\pi$ to be robust to both types requires a wealth of task-specific training data that can be expensive for humans to annotate and produce (Peng et al., 2023). Below, we offer a more efficient way to do so by constructing a task-specific state abstraction.

# 4 LANGUAGE-GUIDED ABSTRACTION (LGA)

Traditional (GC)BC forces the behavioral policy to learn a joint distribution over language, observations, and actions—effectively requiring the robot to develop both language and perceptual scene understanding simultaneously, and to ground language in the current observation. Our approach instead *offloads* contextual language understanding to a LM which identifies task-relevant features in the perceptual state. Unlike other recent methods that offload language understanding to LMs (Ahn et al., 2022), we do not rely on libraries of pre-trained language-annotated skills, which may be insufficient when there are key behaviors indescribable in language. Instead, we introduce a *state abstraction function* that takes the raw perceptual inputs and language-specified goal, and outputs a set of task-relevant features. Intuitively, LGA can be seen as a form of language-guided attention (Bahdanau et al., 2015): it conditions the robot's observations on language, removing the burden of language understanding from the policy. We describe our general framework below, which we will instantiate in Section 5.

## 4.1 STATE ABSTRACTION FUNCTION

Formally, we define a *state abstraction function* $\hat{f}$ that produces task-relevant state representations: $\hat{f} : \mathcal{S} \times \mathcal{L} \to \hat{\mathcal{S}}$, consisting of three steps:

**Textualization** ($s \to \phi$). First, similar to other LM-assisted methods (Huang et al., 2022b; Ahn et al., 2022) the raw perceptual input $s$ is transformed into a text-based feature set $\phi$, representing a set of features (described in natural language) that encapsulate the agent's full perceptual inputs.[1] This text representation may include common visual attributes of the observation like objects in the scene, which are extractable via segmentation models (Kirillov et al., 2023). In Figure 1B, for example, textualization transforms observations to a feature set of object names and pixel locations.

**Feature abstraction** ($\phi \to \hat{\phi}$). Given a feature set $\phi$, we achieve **abstraction** by using an LM to select the subset of features from $\phi$ relevant to the task $\ell$: $(\phi, \ell) \to \hat{\phi}$. In Figure 1B, abstraction removes distractor objects from the feature set, in this case preserving only the target object (*orange*).

**Instantiation** ($\hat{\phi} \to \hat{s}$). As a last step, we transform the abstracted feature set back into an (abstracted) perceptual input with only relevant features on display: $\hat{\phi} \to \hat{s}$. This step is required for rendering the abstracted feature set in a form usable by a policy. In Figure 1B, for example, this step transforms the abstracted feature set into an observation showing only the relevant object.

## 4.2 ABSTRACTION-CONDITIONED POLICY LEARNING

After receiving an abstract state $\hat{s}$ from the state abstraction function, we learn a policy mapping from the abstract state to actions, yielding $\pi_{\hat{\psi}} : \hat{\mathcal{S}} \to \mathcal{A}$.[2] $\pi_{\hat{\psi}}$ is trained to minimize the loss:

$$\mathcal{L}_{\text{LGA}} = \mathbb{E}_{(s_t^i, a_t^i, \ell^i) \sim D_{\text{train}}}[||\pi_{\hat{\psi}}(\hat{f}(s_t^i, \ell^i)) - a_t^i||_2^2], \tag{3}$$

with the differences from GCBC (Eq. 2) highlighted in red. The LGA policy $\pi_{\hat{\psi}}$ never sees the language input $\ell$; instead, it operates over the language-conditioned state representation, $\hat{s}$. Consequently, it can exhibit non-trivial generalization capabilities relative to Eqs. 1 and 2, which given a single goal seen at training and test, collapse to the same non-generalizable policy.

LGA offers several appealing properties relative to traditional (GC)BC. First, it mitigates spurious correlations during training because all goal information is highlighted in semantic maps rather than raw pixels. Second, the LM can help resolve ambiguous goals present at test by converting the language utterance and test observation into an unambiguous abstract state — effective even if LGA has only seen a single unambiguous task at training. Therefore, using LMs confer valuable generalization capabilities at test-time, as LMs can "intercede" to determine only the contextually-appropriate task-relevant features for policy input.

---

[1]In the general case, this could be implemented via segmentation (Kirillov et al., 2023) and a captioner.

[2]LGA can also be used to *augment* the original observation, in which case a policy is learned over the concatenated original and abstracted state-spaces, e.g., $\pi_{\hat{\psi}} : \hat{\mathcal{S}} \times \mathcal{S} \to \mathcal{A}$. This is a more conservative approach, allowing the policy to observe both the original and abstracted states.

## 5 EXPERIMENTAL SETUP

### 5.1 ENVIRONMENT AND FEATURE SPACE

We generate robotic control tasks from VIMA (Jiang et al., 2022), a vision-based manipulation environment. We study three types of tasks: *pick-and-place* of an object onto a goal location, *rotation* of an object to a pre-specified degree, and *sweep* of an object while avoiding a potential obstacle. The environment consists of a large feature space (29 objects, e.g. *bowl*, and 81 colors/textures, e.g. *wooden*) of various semantic meaning. Observations are top-down RGB images and actions are continuous pick-and-place or sweep poses each consisting of a 2D coordinate and a rotation expressed as a quaternion. Success is determined by executing the correct action on the target object within radius $\epsilon$ of the goal.

VIMA is an appealing domain for evaluation due to (1) its combinatorially large feature space for assessing policy generalization and (2) the presence of tasks that require behaviors that are difficult to specify in language (e.g. velocity, rotation, etc.). The latter means we cannot deploy an instruction-following method with a library of language-annotated, pre-trained skills.

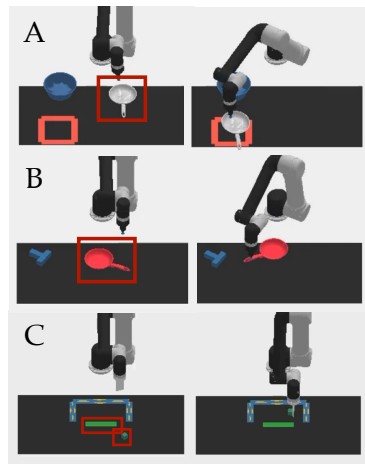

Figure 2: We evaluate on three task settings in VIMA. **A**: Pick-and-place. **B**: Rotate. **C**: Sweep while avoid. Red bounding boxes depict task-relevant features which must be accounted for in the abstraction.

The task-relevant feature set $\hat{\phi}$ is specified via a JSON containing the set of possible task-relevant objects and colors. For **pick-and-place/rotate**, this feature set includes the target object; for **sweep**, this feature set parameterizes both the target object and the object to be avoided. Because the feature space can be combinatorially large, constructing abstractions that preserve only task-relevant features of an ambiguously specified objective, e.g. *something that holds water*, is challenging.

### 5.2 LGA IMPLEMENTATION

**Textualization.** We first extract a structured feature set $\phi$ by obtaining the ground truth state segmentation mask and object descriptions from the simulator.

**Feature abstraction.** We use two versions of LGA for defining relevant feature sets $\hat{\phi}$. **LGA** uses GPT4 (OpenAI, 2023) to specify a relevant feature set.[3] We give GPT4 a description of the VIMA environment (including a list of all possible object types and colors), the goal utterance, and a target object type or feature to evaluate (full prompt and additional details can be found in appendix A.2). GPT4 provides a binary response indicating whether the object type or feature should be included in the abstraction. **LGA-HILL** extends this by placing a human in the loop. The human is shown the proposed abstraction and can refine it by adding or removing features (see Section 5.4 for details).

**Instantiation.** As described in Section 5.1, the abstract state representation consists of a distribution over *object types* and *object colors*, including both target objects and relevant obstacles. Given a specified task-relevant object type and color, we use an image editor to identify all salient objects in the observation. In particular, the image editor produces a "goal mask", a pixel mask where the relevant objects are converted to ones and everything else is zeroed out. [4] In our setting, the simulator provides the ground-truth scene segmentation which we use to produce the goal mask $\hat{s}$.

**Abstraction-Augmented Policy Learning** We instantiate the abstraction-augmented policy procedure, described in Section 4.2, in our environment. We assume access to ground truth demonstrations generated by an oracle. LGA learns a policy from scratch via imitation learning on these demonstrations, using $\hat{s}$. We implement a CNN architecture that processes abstract states into embeddings, which we then feed through a MLP for action prediction. See Appendix A.3 for details.

---

[3]We use the `gpt-4-0613` version of GPT4.

[4]Here, we produce a simple binary pixel mask highlighting relevant objects. In general, however, it may be fruitful to experiment with more general scalar cost maps, such as highlighting areas that should be avoided.

## 5.3 Training and Test Tasks

**Training Tasks.** We are interested in studying how well different methods perform task-relevant feature selection, i.e. specify $\hat{\phi}$ given a single language description of the task. We evaluate the quality of abstractions by using them to construct a *training distribution* for each task (on which we perform imitation learning). We want methods which construct a *minimal* abstraction $\hat{\phi}$ that *contains features necessary* for the task. If these abstractions do not contain all features necessary for completion of the task, then the policy will not succeed; if these abstractions are not minimal, then learning will be less data-efficient. Thus, we evaluate both *data efficiency* and *task performance*; good abstractions will result in the policy achieving higher success with fewer demonstrations.

We construct this distribution by placing a uniform probability over the set of task-relevant features specified in $\hat{\phi}$. To create a task, we sample an object and texture from this distribution as the "target" object and place it randomly in the environment. For example, for *something that can hold water*, we sample an object from {*pan, bowl, container*} and sample a color from the universe of all colors. In the *sweep* task, we perform this same process, but for possible obstacles. We then generate a random distractor object. Distractors, target objects, and obstacles are randomly placed in one of four discretized state locations. Last, for pick-and-place, we place a fixed goal object (pallet) in the state as the "place" goal. We generate corresponding training demonstrations via an oracle.

**Test Tasks.** In practice, we do not have access to ground-truth distributions – this specification must come from the end user. However, for our experiments, the first three authors defined "true" distributions from which we sampled starting states for each task. See details in Appendix A.2.

## 5.4 Comparisons

We analyze our approach (**LGA** and **LGA-HILL**) against three baselines and two ablations.

**Human (Baseline).** The first baseline explores having users (instead of LMs) manually specify task-relevant features $\hat{\phi}$ in the feature abstraction stage, to isolate the effect of LM-aided specification. All other components remain the same as in LGA. The features are fed to the same CNN architecture as for LGA and the policy is again learned over the generated state abstractions.

**GCBC-DART (Baseline)**: Our second baseline is (goal-conditioned) behavior cloning, described in Eq. 2. Because imitation learning is notoriously brittle to distribution shift, we implement a stronger variant (DART) (Laskey et al., 2017) by injecting Gaussian noise (swept from $\mu = 0.0$ to $0.2$, $\beta = 0.5$) into the collected demonstrations, sampling 5 noisy demonstrations per true demonstration. We implement goal-conditioning by concatenating an LM embedding of the goal utterance with a CNN embedding of the state and then learning a policy over the joint representation. We generate language embeddings using Sentence-BERT (Reimers & Gurevych, 2019).

**GCBC-SEG (Baseline):** To study the impact of object segmentation, our third baseline implements goal-conditioning on the segmentation mask in addition to the observation. We independently process the two observations via CNNs, and concatenate the embeddings as a joint representation.

**LGA-S (Ablation):** We explore the effect of training with both the raw observation $s$ in addition to $\hat{s}$ to study the impact of discarding the original observation. The two observations are independently processed via CNNs, then concatenated as a joint representation.

**LGA-L (Ablation):** We implement a variant of abstraction where we condition only on the language embedding of the relevant features $\hat{\phi}$ rather than on the state abstraction $\hat{s}$.

**Human Data Collection.** For comparisons that require human data (**LGA-HILL** and **Human**), we conducted a computer-based in-person user study to assess the ability of humans to specify task-relevant features $\hat{\phi}$ in the *feature abstraction* step, both with and without an LM. We recruited 18 participants (67% male, aged 22-44) from the greater MIT community. We paid each participant $30 for their time. Our study received institutional IRB approval and all user data was anonymized.

We first introduce the user to the environment and full feature space $\phi$. To additionally help participants, we provide both a text and visual representation of features (the latter of which the LM does not see). We also walk the user through an example task specification. In the **Human** condition, we introduce task objectives (detailed in Section 5) sequentially to the user and ask for them to specify

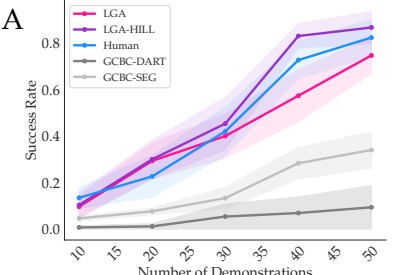 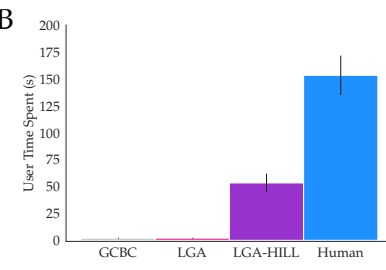

Figure 3: **(Q1) A**: Comparing task performance (averaged over all tasks) of each method when controlling the number of training demonstrations. **B**: Comparing the amount of time (averaged over all tasks) that human users spent specifying task-relevant features for each method. LGA outperforms baselines on task performance while significantly reducing user time spent compared to manual feature specification ($p < 0.001$).

$\hat{\phi}$ by typing their selected features into a task design file (we provide the full feature list for easy access). In the **LGA-HILL** condition, we show the same objectives, but with LM answers prefilled as the "raw" $\hat{\phi}$. To minimize ordering bias, we randomly assign half of our participants to begin with each condition. For both conditions, we measure the time each user spent specifying the feature set.

## 6 RESULTS

### 6.1 Q1: IMPROVING EASE AND POLICY PERFORMANCE OF TASK SPECIFICATION

We begin by evaluating both LGA and LGA-HILL against the baseline methods on nine scenarios from pick-and-place, two from rotate, and two from sweep. Each scenario demonstrates varied feature complexity, where the ground truth task distributions may be combinatorially large. Each method learns a single-task policy from scratch for each of the tasks.

**Metrics.** We report two metrics: *task performance vs. # of demonstrations* and *user time*. Task performance is assessed as the success rate of the policy on 20 sampled test states. We plot this with respect to the number of training demonstrations an agent sees. User specification time is measured as the time (in seconds) each user spent specifying the task-relevant feature set for each task.

**Task Objectives.** For pick-and-place, we construct nine scenarios with a wide range of target objects: *red heart*, *heart*, *letter*, *tiger-colored object*, *letter from the word letter*, *consonant with a warm color*, *vowel with multiple colors*, *something to drink water out of*, and *something from a typical kitchen*. We chose these tasks to 1) test the ability of the LM in LGA to semantically reason over which target objects and features satisfy human properties, e.g., *to drink water out of*, and 2) explore how potentially laborious performing manual feature specification from large feature sets can be for human users. For rotate, we construct two scenarios: *rotate block*, *rotate something I can put food in*, to illustrate LGA's ability to learn behaviors that are difficult to specify in language (e.g., rotate an object 125 degrees), but easy to specify for task-relevance (e.g., pan). For sweep, we construct two scenarios: *sweep the block without touching the pan* and *sweep the block without touching the line* to illustrate LGA's ability to identify task-relevant features that are both target objects and obstacles.

**Results.** We vary the number of demonstrations that each policy is trained on from 10 to 50. We visualize the resulting policy performance in Figure 3 (A). In Figure 3 (B), we visualize how much time users spent specifying the task-relevant features for each method.[5] From these results, it is clear that the feature-specification methods (i.e. LGA, LGA-HILL, and Human) are more sample-efficient than baselines with consistently higher performance at each number of training demonstrations. Furthermore, LGA methods require significantly less user time than the Human baseline ($p < 0.001$ using a paired t-test for all tasks). We note that despite comparable task performance between LGA and LGA-HILL, the latter remains a valuable instantiation for tasks that require human input, such as high-stakes decision-making scenarios or personalization to individual user preferences.

***Summary.*** *LGA requires significantly less human time than manually specifying features, but still leads to more sample-efficient performance compared to baselines.*

---

[5]While we assign zero time spent specifying features for GCBC in our experiments, in practice, user time would be instead spent specifying initial state configurations for generating every demonstration, which only disadvantages naive GCBC even more when compared to methods that specify features.

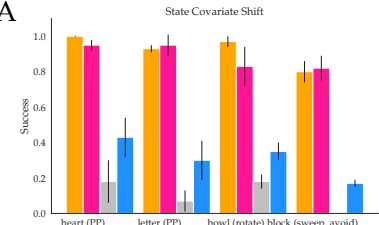 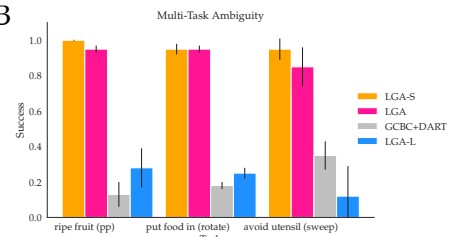

Figure 4: **(Q2) A**: Results on state covariate shifts. LGA variants that condition on state abstractions (with or without the original observation) outperform LGA-L (which conditions on the language abstraction only) and GCBC+DART (which attempts to use noise injection to handle covariate shift). **(Q3) B**: Results on multi-task ambiguity. We observe the same trends, with LGA variants that condition on state abstractions able to resolve previously unseen linguistic commands.

## 6.2 Q2: IMPROVING POLICY ROBUSTNESS TO OBSERVATIONAL COVARIATE SHIFT

We have shown that LGA enables faster feature specification when compared to hand-engineering features from scratch, and also leads to more sample-efficient policies when compared to traditional imitation learning. We now ask how the different design choices taken in LGA handle state covariate shift compared to GCBC+DART, which is a stronger imitation learning baseline designed to help mitigate covariate shift by synthetically broadening the demonstrations data with injected noise.

**Task Objectives.** We evaluate on four previously defined scenarios in Q1: *heart* (pick-and-place), *letter* (pick-and-place), *bowl* (rotate), and *block* (sweep while avoid). For each task, we now construct state covariate shifts designed to test policy robustness.

For all tasks, we define a training distribution where the task-relevant features contain some subset of the feature list, e.g., *red* and *blue*. At test time, we evaluate by sampling tasks from a distribution where either 1) task-relevant textures change (e.g., objects are now *pink*) or 2) a sampled distractor is added to the scene. These tasks are intended to evaluate LGA's flexibility in including (and excluding) appropriate task-relevant features relative to imitation learning over raw observations.

**Results.** For each method, we train on 50 demonstrations from the train distribution then evaluate on 20 tasks from the test distribution. As shown in Fig. 4A, policies trained with either of the LGA methods are more robust to state covariate shifts than those trained with GCBC+DART. Notably, we observe that LGA-L (our ablation where we provide the language abstraction only) underperforms the variants that operate over state abstractions. This makes sense considering observations (even if masked) contain information that directly grounds task-relevant information for the policy to operate over into the environment. This confirms our hypothesis that providing a non-ambiguous state abstraction that already precludes non-relevant task features from the state result in policies that are less brittle to spuriously correlated state features.

***Summary.*** *Policies trained with LGA state abstractions (with or without raw observations) are more robust to observational shift than GCBC+DART.*

## 6.3 Q3: IMPROVING MULTI-TASK POLICY ROBUSTNESS TO LINGUISTIC AMBIGUITY

The previous experiments have focused on evaluating LGA in single-task settings where each policy corresponds to a singular language specification. We now evaluate how LGA performs in the multi-task case where test time environments allow for multiple task specifications, and conditioning on language is necessary to resolve the ambiguity.

**Task Objectives.** We now define three new tasks: *pick up fruit* (pick-and-place), *put food in* (rotate), and *avoid utensil* (sweep). For each task, we train a multi-task policy. We define our training distribution as states that contain a single object from each specification (e.g., a *tomato* or *apple*). At test time, we present the policy with states that now contain both objects. We condition the policy on both the linguistic utterances it has seen before: e.g., "Bring me a tomato." or "Bring me an apple", and as well as one it has not: e.g., "Bring me a fruit." We now evaluate LGA's zero-shot generalization performance on both seen and unseen linguistic goal utterances.

**Results.** For each method, we train on 50 demonstrations sampled from the train distribution for each task, then evaluate on 20 task instances from the test distribution. We visualize the averaged

results in Figure 4B. These results demonstrate LGA's ability to flexibly construct state abstractions *even if the language utterance was previously unseen.*

***Summary.*** *Multi-task policies trained with LGA can better resolve task ambiguity compared to GCBC+DART as they can flexibly adapt new language specifications to the observation.*

# 7 REAL-WORLD ROBOTICS TASKS

We highlight LGA's real world applicability on a Spot robot performing mobile manipulation tasks.

**Robot Platform.** Spot [6] is a mobile manipulation legged robot equipped with six RGB-D cameras (one in gripper, two in front, one on each side, one in back), each producing an observation of size $480 \times 640$. For our demonstrations, we only use observations taken from the robot's front camera.

**Tasks and Data Collection.** We collected demonstrations of a human operator teleoperating the robot while performing two mobile manipulation tasks with various household objects: *bring fruit* and *throw away object*. The manipulation action space consists of the following three actions along with their parameters: (*xy, grasp*), (*xy, move*), (*drop*) while the navigation action space consists of a SE(3) group denoting navigation waypoints. For *bring fruit*, the robot is tasked with picking up fruit on a table, bringing it to a user at a specified location, and dropping it in their hand. For *throw away drink*, the robot is tasked with picking up a drink from a table, moving to a specified location, and dropping it into a recycling bin. Both tasks include possible distractors like brushes and drills on the table. For each task, we collected a single demonstration of the robot successfully performing the task (e.g., throwing away a soda can). The full action sequence is recorded for training.

**Training and Test Procedure.** First, we extract a segmented image using Segment Anything (Kirillov et al., 2023) and captioner Dedic (Zhou et al., 2022); second, we query the feature list along with the linguistic specification (e.g., *throw away drink*) to construct our abstract feature list; last, we map the abstract feature list back into the original observation as an abstracted state. We train a policy to map from the abstracted state into the action sequence.[7] For test, we then change the scene (e.g., with a new target object like a water bottle instead of a soda can, along with distractors) and predict the action sequence from the new observation.

**Takeaway.** LGA produced policies capable of successfully completing both tasks consistently. The failures we did observe were largely due to captioning errors (e.g., the segmentation model detected the object but was unable to produce a good text description).

# 8 DISCUSSION & CONCLUSION

We proposed LGA to leverage LMs to learn generalizable policies on state abstractions produced for task-relevant features. Our results confirmed that LGA produces state abstractions similarly effective to those manually designed by human annotators while requiring significantly less user time than manual specification. Furthermore, we showed that these abstractions yield policies robust to observational covariate shift and ambiguous language in both single-task and multi-task settings.

**Limitations.** We assume state abstractions for training good policies can be captured by visual features present in individual scenes. An exciting direction for future work would be to learn how to build state abstractions for trajectory-relevant features. We also assume features are a) segmentable and b) expressible in language, i.e. a LM (or human) can understand how each feature may or may not relate to the task. In more complex scenarios, many real-world tasks may be dependent on features that are less expressible in text, e.g. relative position, and are difficult to encapsulate in an abstraction. Although there is evidence to suggest even ungrounded LMs can learn grounded concepts such as spatial relations (Patel & Pavlick, 2021), we did not rigorously test this. We are excited to explore additional feature representations (perhaps multimodal visualization interfaces in addition to language representations) that can be jointly used to construct state abstractions.

---

[6]Our Spot's name is Moana.

[7]For latency purposes and ease of data generation, we perform imitation learning over the full trajectory rather than each state (i.e. predict a sequence of actions from an initial observation only).

## 9 ACKNOWLEDGEMENTS

We thank the Boston Dynamics AI Institute for providing robotic hardware resources used in this project, as well as Nishanth Kumar for help in running (late-night) demonstration videos. We thank members of the Language and Intelligence and Interactive Robotics Groups for helpful feedback and discussions. Andi Peng is supported by the NSF Graduate Research Fellowship and Open Philanthropy. Belinda Li and Theodore Sumers are supported by National Defense Science and Engineering Graduate Fellowships. Ilia Sucholutsky is supported by a NSERC Fellowship (567554-2022). This work was supported by the National Science Foundation under grant IIS-2238240.

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

# A  APPENDIX

## A.1  ENVIRONMENT DETAILS

States are fully-observable images and represent a RGB fixed-camera topdown view of the scene. The action space is continuous of dimension 4 and consists of high-level pick-and-place actions parameterized by the pose of the end effector. There are 4 possible objects that can be spawned: the target (manipulated) object, goal object, and (up to) two possible distractors. There are two types of target features (object type and texture) and 29 possible instantiations of object type and 81 instantiations of texture. Visual depictions of the features can be found in the user study below.

## A.2  TASK DETAILS

We provide details regarding all tasks, including ground truth task distributions and full LM prompt.

**Pick-and-place:**

*red heart*:

- Task prompt: *Bring me the red heart.*
- True distribution: {objects: heart}, {textures: red, dark red, dark red swirl, red paisley}

*heart*:

- Task prompt: *Bring me the heart.*
- True distribution: {objects: heart}, {textures: ALL}

*tiger-colored object*:

- Task prompt: *Bring me the tiger-colored object.*
- True distribution: {objects: ALL}, {textures: tiger}

*letter from the word letter*:

- Task prompt: *Bring me a letter from the word 'letter'.*
- True distribution: {objects: letter E, letter R, letter T}, {textures: ALL}

*consonant with warm-color*:

- Task prompt: *Bring me a consonant with a warm color on it.*
- True distribution: {objects: letter G, letter M, letter R, letter T, letter V}, {textures: dark {red—yellow—pink—orange}, dark {red—yellow—pink—orange} and * stripe, {red—yellow—pink—orange} and * polka dot, dark {red—yellow—pink—orange} and * polka dot, {red—yellow—pink} swirl, dark {red—yellow—pink} swirl, {red—yellow—pink} paisley, tiger, magma, wooden, rainbow, tiles, brick}

*vowel with multiple colors*:

- Task prompt: *Bring me a vowel with multiple colors on it.*
- True distribution: {objects: letter A, letter E}, {textures: polka dot, tiles, checkerboard, plastic, tiger, magma, rainbow, * and * stripe, * and * polka dot, * swirl, * paisley}

*something to drink water out of*:

- Task prompt: *Bring me something to drink water out of.*
- True distribution: {objects: bowl, pan, container}, {textures: ALL}

*something from a typical kitchen*:

- Task prompt: *Bring me something from a typical kitchen.*
- True distribution: {objects: bowl, pan, container}, {textures: ALL}

**Rotate:**

*block*:

- Task prompt: *Rotate the block.*

- True distribution: {objects: block, small block, shorter block, L shaped block}, {textures: ALL}

*something to drink water out of*:

- Task prompt: *Rotate something to drink water out of.*
- True distribution: {objects: container, bowl, pan}, {textures: red, dark red, dark red swirl, red paisley}

**Sweep While Avoid:**

*block, pan*:

- Task prompt: *Sweep the block without touching the pan.*
- True distribution: {objects: block, small block, shorter block, L shaped block, pan}, {textures: ALL}

*block, line*:

- Task prompt: *Sweep the block without touching the line.*
- True distribution: {objects: block, small block, shorter block, L shaped block, line}, {textures: ALL}

### A.2.1 FULL PROMPT

ChatGPT models (including GPT4) can take in both system prompts and user prompts. We split our prompt into these two parts as follows.

System prompt where {object_list} is replaced by the list of all object types in the environment and {object_colors} by the list of all colors and textures:

> You are interfacing with a robotics environment that has a robotic arm learning to manipulate objects based on some linguistic command (e.g. "pick up the red bowl"). At each interaction, the researcher will specify the command that you need to teach the robot. In order to teach the robot, you will need to help design the training distribution by specifying what properties task-relevant objects can have based on the given command. Objects in this environment have two properties: object type, object color. Any object type can be paired with any color, but an object can only take on exactly one object type and exactly one color.
> Object types:
> {object_list}
> Object colors:
> {object_colors}

User prompt where {rule} is replaced by one of the task prompts listed above, {group} is replaced by "object color" or "object type", and {candidate} is replaced by each candidate object color or type that we would like the LM to evaluate:

> The command is "{rule}". In an instantiation of the environment that contains only some subset of the object types and colors, could the target object have {group} "{candidate}"? Think step-by-step and then finish with a new line that says "Final answer:" followed by "yes" or "no".

### A.3 ARCHITECTURE AND TRAINING DETAILS

**Architecture.**

For GCBC, Sentence-BERT processes a goal utterance in natural language into an embedding of size 384, which we additionally process through a linear MLP of output size 100. We process the state using a standard Conv2D block consisting of 3 stacked Conv2D layers of output channel sizes 32, 64, 32, and strides 4, 2, and 1. Each output layer is processed by BatchNorm2D as well as a ReLU activation. After the last Conv2D layer, we flatten the output and concatenate with the output of the goal utterance. We feed the concatenated output through a last linear layer for action prediction.

For LGA, we process both the state and the state abstraction through dual Conv2Ds blocks like above. We concatenate the output and feed through a last linear layer for action prediction.

**Training.**

We train all networks to convergence for a maximum of 750 epochs. All computation was done on two NVIDIA GeForce RTX 3090 GPUs. Rollouts were rendered locally using PyBullet on commercial hardware (a Macbook Pro).

## A.4 USER STUDY

In the following pages, we have included the full user study shared with participants. Following standard user study procedure, we initial briefed users by telling them how long the study was (roughly 30 mins) and that they were free to leave at anytime. Demographic information was collected in person. We showed participants the first pdf during the familiarization phase to introduce them to the environment and full feature list. We then randomly chose to begin with either the human-only task (second pdf) or the LM-aided task (third pdf). After the study, users were debriefed and given the email of the study designer to contact if they had any questions.

# User Study: Instructions and Dictionary

You are helping teach a robot which properties are important to a desired task.

The robot is learning to pick up objects based on some linguistic command (e.g. "pick up red bowl").

At each interaction, we will specify the task that you need to teach the robot. In order to teach the robot, you will need to help design the task by specifying what properties the target object can have based on the given command.

Target objects in this environment have two properties: [*object type, object color*].

Note: Any object type can be paired with any color, but an object can only take on *exactly one object type and exactly one color*.

Object types:

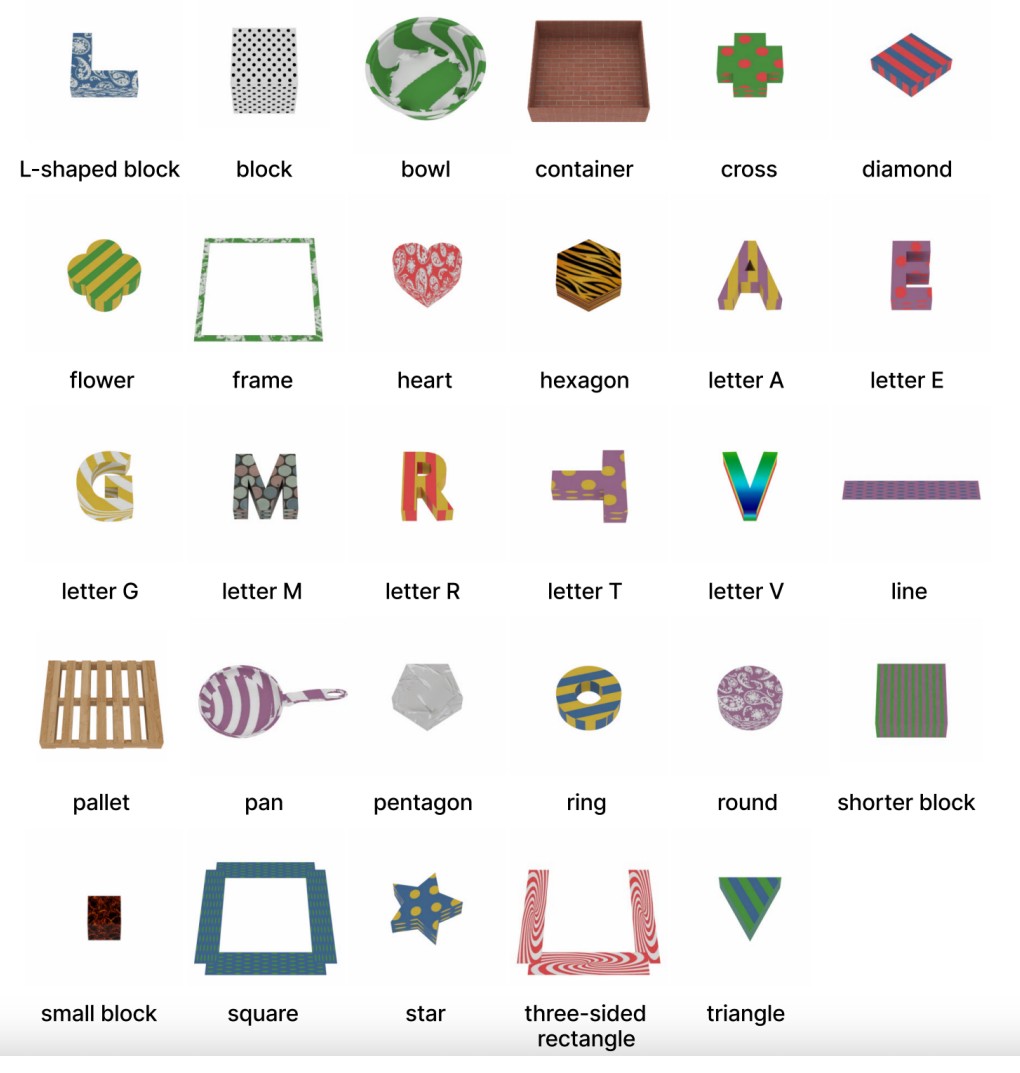

| L-shaped block | block | bowl | container | cross | diamond |
| flower | frame | heart | hexagon | letter A | letter E |
| letter G | letter M | letter R | letter T | letter V | line |
| pallet | pan | pentagon | ring | round | shorter block |
| small block | square | star | three-sided rectangle | triangle | |

## Object colors:

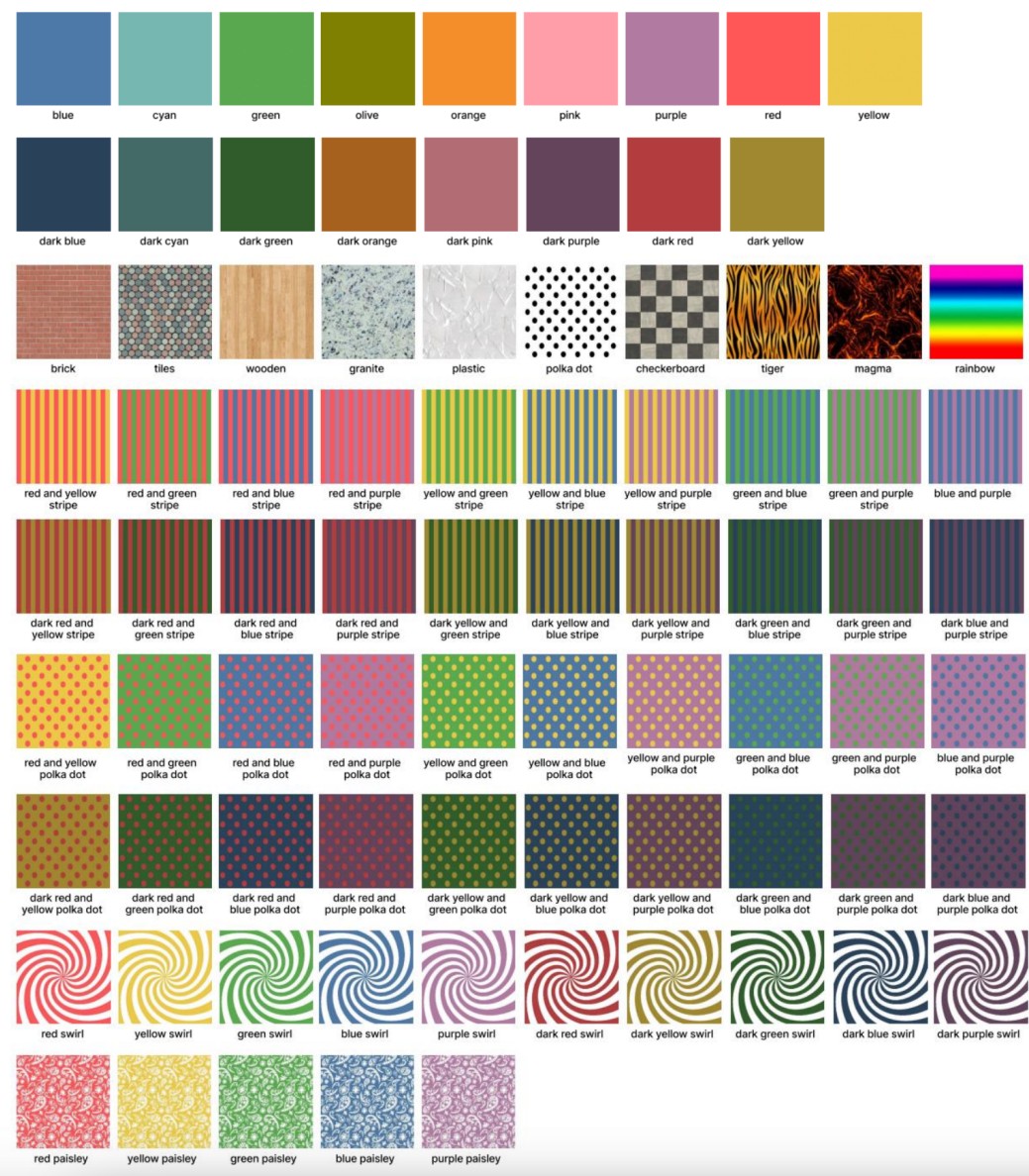

## Example task: "Bring me polka dot geometric shape"

```python
In [3]: task_kwargs = {
            'possible_dragged_obj': ['diamond',
                                     'triangle',
                                     'hexagon',
                                     'pentagon',
                                     'square'],
            'possible_dragged_obj_texture': ['red and yellow polka dot',
                                             'red and green polka dot',
                                             'red and blue polka dot',
                                             'red and purple polka dot',
                                             'yellow and green polka dot',
                                             'yellow and blue polka dot',
                                             'yellow and purple polka dot',
                                             'green and blue polka dot',
                                             'green and purple polka dot',
                                             'blue and purple polka dot',
                                             'dark red and yellow polka dot',
```

```
                                     'dark red and green polka dot',
                                     'dark red and blue polka dot',
                                     'dark red and purple polka dot',
                                     'dark yellow and green polka dot',
                                     'dark yellow and blue polka dot',
                                     'dark yellow and purple polka dot',
                                     'dark green and blue polka dot',
                                     'dark green and purple polka dot',
                                     'dark blue and purple polka dot',]}
```

# Experiment

For your reference, here are the object types and colors (feel free to copy + paste):

In [2]:
```
'possible_dragged_obj': ['L-shaped block',
                         'block',
                         'bowl',
                         'container',
                         'cross',
                         'diamond',
                         'flower',
                         'frame',
                         'heart',
                         'hexagon',
                         'letter A',
                         'letter E',
                         'letter G',
                         'letter M',
                         'letter R',
                         'letter T',
                         'letter V',
                         'line',
                         'pallet',
                         'pan',
                         'pentagon',
                         'ring',
                         'round',
                         'shorter block',
                         'small block',
                         'square',
                         'star',
                         'three-sided rectangle',
                         'triangle']
```

In [ ]:
```
'possible_dragged_obj_texture': ['brick',
                                 'tiles',
                                 'wooden',
                                 'granite',
                                 'plastic',
                                 'polka dot',
                                 'checkerboard',
                                 'tiger',
                                 'magma',
                                 'rainbow',
                                 'blue',
                                 'cyan',
                                 'green',
                                 'olive',
                                 'orange',
                                 'pink',
                                 'purple',
                                 'red',
                                 'yellow',
                                 'dark blue',
                                 'dark cyan',
                                 'dark green',
                                 'dark olive',
                                 'dark orange',
                                 'dark pink',
```

```
                                    'dark purple',
                                    'dark red',
                                    'dark yellow',
                                    'red and yellow stripe',
                                    'red and green stripe',
                                    'red and blue stripe',
                                    'red and purple stripe',
                                    'yellow and green stripe',
                                    'yellow and blue stripe',
                                    'yellow and purple stripe',
                                    'green and blue stripe',
                                    'green and purple stripe',
                                    'blue and purple stripe',
                                    'dark red and yellow stripe',
                                    'dark red and green stripe',
                                    'dark red and blue stripe',
                                    'dark red and purple stripe',
                                    'dark yellow and green stripe',
                                    'dark yellow and blue stripe',
                                    'dark yellow and purple stripe',
                                    'dark green and blue stripe',
                                    'dark green and purple stripe',
                                    'dark blue and purple stripe',
                                    'red and yellow polka dot',
                                    'red and green polka dot',
                                    'red and blue polka dot',
                                    'red and purple polka dot',
                                    'yellow and green polka dot',
                                    'yellow and blue polka dot',
                                    'yellow and purple polka dot',
                                    'green and blue polka dot',
                                    'green and purple polka dot',
                                    'blue and purple polka dot',
                                    'dark red and yellow polka dot',
                                    'dark red and green polka dot',
                                    'dark red and blue polka dot',
                                    'dark red and purple polka dot',
                                    'dark yellow and green polka dot',
                                    'dark yellow and blue polka dot',
                                    'dark yellow and purple polka dot',
                                    'dark green and blue polka dot',
                                    'dark green and purple polka dot',
                                    'dark blue and purple polka dot',
                                    'red swirl',
                                    'yellow swirl',
                                    'green swirl',
                                    'blue swirl',
                                    'purple swirl',
                                    'dark red swirl',
                                    'dark yellow swirl',
                                    'dark green swirl',
                                    'dark blue swirl',
                                    'dark purple swirl',
                                    'red paisley',
                                    'yellow paisley',
                                    'green paisley',
                                    'blue paisley',
                                    'purple paisley']
```

## Task 1: "Bring me the red heart"

```
In [4]:  task_kwargs = {
             'possible_dragged_obj': [],
```

```
        'possible_dragged_obj_texture': []}
```

## Task 2: "Bring me the heart"

```
In [ ]:   task_kwargs = {
              'possible_dragged_obj': [],
              'possible_dragged_obj_texture': []}
```

## Task 3: "Bring me the tiger object"

```
In [ ]:   task_kwargs = {
              'possible_dragged_obj': [],
              'possible_dragged_obj_texture': []}
```

## Task 4: "Bring me a letter from the word letter"

```
In [ ]:   task_kwargs = {
              'possible_dragged_obj': [],
              'possible_dragged_obj_texture': []}
```

## Task 5: "Bring me a consonant that has any warm color on it"

```
In [ ]:   task_kwargs = {
              'possible_dragged_obj': [],
              'possible_dragged_obj_texture': []}
```

## Task 6: "Bring me a vowel that has multiple colors on it" (white is not a color)

```
In [ ]:   task_kwargs = {
              'possible_dragged_obj': [],
              'possible_dragged_obj_texture': []}
```

## Task 7: "Bring me a letter. If there are multiple, bring the one that comes earliest in the alphabet."

```
In [ ]:   task_kwargs = {
              'possible_dragged_obj': [],
              'possible_dragged_obj_texture': []}
```

## Task 8: "Bring me something I can drink water out of"

```
In [ ]:   task_kwargs = {
              'possible_dragged_obj': [],
              'possible_dragged_obj_texture': []}
```

## Task 9: "Bring me something I can find in a typical kitchen"

```
In [ ]:   task_kwargs = {
              'possible_dragged_obj': [],
              'possible_dragged_obj_texture': []}
```

## Experiment

For your reference, here are the object types and colors (feel free to copy + paste).

A large language model (GPT-4) has given its answers to help you with the task (they are pre-filled in). You may feel free to change/modify the answers, or accept them without change:

```
In [ ]:  'possible_dragged_obj': ['L-shaped block',
                                   'block',
                                   'bowl',
                                   'container',
                                   'cross',
                                   'diamond',
                                   'flower',
                                   'frame',
                                   'heart',
                                   'hexagon',
                                   'letter A',
                                   'letter E',
                                   'letter G',
                                   'letter M',
                                   'letter R',
                                   'letter T',
                                   'letter V',
                                   'line',
                                   'pallet',
                                   'pan',
                                   'pentagon',
                                   'ring',
                                   'round',
                                   'shorter block',
                                   'small block',
                                   'square',
                                   'star',
                                   'three-sided rectangle',
                                   'triangle']
```

```
In [ ]:  'possible_dragged_obj_texture': ['brick',
                                           'tiles',
                                           'wooden',
                                           'granite',
                                           'plastic',
                                           'polka dot',
                                           'checkerboard',
                                           'tiger',
                                           'magma',
                                           'rainbow',
                                           'blue',
                                           'cyan',
                                           'green',
                                           'olive',
                                           'orange',
                                           'pink',
                                           'purple',
                                           'red',
                                           'yellow',
                                           'dark blue',
                                           'dark cyan',
```

```
                                    'dark green',
                                    'dark olive',
                                    'dark orange',
                                    'dark pink',
                                    'dark purple',
                                    'dark red',
                                    'dark yellow',
                                    'red and yellow stripe',
                                    'red and green stripe',
                                    'red and blue stripe',
                                    'red and purple stripe',
                                    'yellow and green stripe',
                                    'yellow and blue stripe',
                                    'yellow and purple stripe',
                                    'green and blue stripe',
                                    'green and purple stripe',
                                    'blue and purple stripe',
                                    'dark red and yellow stripe',
                                    'dark red and green stripe',
                                    'dark red and blue stripe',
                                    'dark red and purple stripe',
                                    'dark yellow and green stripe',
                                    'dark yellow and blue stripe',
                                    'dark yellow and purple stripe',
                                    'dark green and blue stripe',
                                    'dark green and purple stripe',
                                    'dark blue and purple stripe',
                                    'red and yellow polka dot',
                                    'red and green polka dot',
                                    'red and blue polka dot',
                                    'red and purple polka dot',
                                    'yellow and green polka dot',
                                    'yellow and blue polka dot',
                                    'yellow and purple polka dot',
                                    'green and blue polka dot',
                                    'green and purple polka dot',
                                    'blue and purple polka dot',
                                    'dark red and yellow polka dot',
                                    'dark red and green polka dot',
                                    'dark red and blue polka dot',
                                    'dark red and purple polka dot',
                                    'dark yellow and green polka dot',
                                    'dark yellow and blue polka dot',
                                    'dark yellow and purple polka dot',
                                    'dark green and blue polka dot',
                                    'dark green and purple polka dot',
                                    'dark blue and purple polka dot',
                                    'red swirl',
                                    'yellow swirl',
                                    'green swirl',
                                    'blue swirl',
                                    'purple swirl',
                                    'dark red swirl',
                                    'dark yellow swirl',
                                    'dark green swirl',
                                    'dark blue swirl',
                                    'dark purple swirl',
                                    'red paisley',
                                    'yellow paisley',
                                    'green paisley',
                                    'blue paisley',
                                    'purple paisley']
```

## Task 1: "Bring me the red heart"

```
task_kwargs = {
    'possible_dragged_obj': ['heart'],
    'possible_dragged_obj_texture': ['red',
                                     'dark red',
                                     'red swirl',
                                     'dark red swirl'
                                     'red paisley']}
```

## Task 2: "Bring me the heart"

```
task_kwargs = {
    'possible_dragged_obj': ['heart'],
    'possible_dragged_obj_texture': ['brick',
                                     'tiles',
                                     'wooden',
                                     'granite',
                                     'plastic',
                                     'polka dot',
                                     'checkerboard',
                                     'tiger',
                                     'magma',
                                     'rainbow',
                                     'blue',
                                     'cyan',
                                     'green',
                                     'olive',
                                     'orange',
                                     'pink',
                                     'purple',
                                     'red',
                                     'yellow',
                                     'dark blue',
                                     'dark cyan',
                                     'dark green',
                                     'dark olive',
                                     'dark orange',
                                     'dark pink',
                                     'dark purple',
                                     'dark red',
                                     'dark yellow',
                                     'red and yellow stripe',
                                     'red and green stripe',
                                     'red and blue stripe',
                                     'red and purple stripe',
                                     'yellow and green stripe',
                                     'yellow and blue stripe',
                                     'yellow and purple stripe',
                                     'green and blue stripe',
                                     'green and purple stripe',
                                     'blue and purple stripe',
                                     'dark red and yellow stripe',
                                     'dark red and green stripe',
                                     'dark red and blue stripe',
                                     'dark red and purple stripe',
                                     'dark yellow and green stripe',
                                     'dark yellow and blue stripe',
                                     'dark yellow and purple stripe',
                                     'dark green and blue stripe',
                                     'dark green and purple stripe',
                                     'dark blue and purple stripe',
```

```
                                    'red and yellow polka dot',
                                    'red and green polka dot',
                                    'red and blue polka dot',
                                    'red and purple polka dot',
                                    'yellow and green polka dot',
                                    'yellow and blue polka dot',
                                    'yellow and purple polka dot',
                                    'green and blue polka dot',
                                    'green and purple polka dot',
                                    'blue and purple polka dot',
                                    'dark red and yellow polka dot',
                                    'dark red and green polka dot',
                                    'dark red and blue polka dot',
                                    'dark red and purple polka dot',
                                    'dark yellow and green polka dot',
                                    'dark yellow and blue polka dot',
                                    'dark yellow and purple polka dot',
                                    'dark green and blue polka dot',
                                    'dark green and purple polka dot',
                                    'dark blue and purple polka dot',
                                    'red swirl',
                                    'yellow swirl',
                                    'green swirl',
                                    'blue swirl',
                                    'purple swirl',
                                    'dark red swirl',
                                    'dark yellow swirl',
                                    'dark green swirl',
                                    'dark blue swirl',
                                    'dark purple swirl',
                                    'red paisley',
                                    'yellow paisley',
                                    'green paisley',
                                    'blue paisley',
                                    'purple paisley']}
```

## Task 3: "Bring me the tiger object"

```
In [ ]:  task_kwargs = {
             'possible_dragged_obj': ['L-shaped block',
                              'block',
                              'bowl',
                              'container',
                              'cross',
                              'diamond',
                              'flower',
                              'frame',
                              'heart',
                              'hexagon',
                              'letter A',
                              'letter E',
                              'letter G',
                              'letter M',
                              'letter R',
                              'letter T',
                              'letter V',
                              'line',
                              'pallet',
                              'pan',
                              'pentagon',
                              'ring',
                              'round',
                              'shorter block',
```

```
                                  'small block',
                                  'square',
                                  'star',
                                  'three-sided rectangle',
                                  'triangle'],
            'possible_dragged_obj_texture': ['tiger']}
```

## Task 4: "Bring me a letter from the word letter"

```
In [ ]: task_kwargs = {
            'possible_dragged_obj': ['letter E'
                                     'letter R',
                                     'letter T'],
            'possible_dragged_obj_texture': ['brick',
                                             'tiles',
                                             'wooden',
                                             'granite',
                                             'plastic',
                                             'polka dot',
                                             'checkerboard',
                                             'tiger',
                                             'magma',
                                             'rainbow',
                                             'blue',
                                             'cyan',
                                             'green',
                                             'olive',
                                             'orange',
                                             'pink',
                                             'purple',
                                             'red',
                                             'yellow',
                                             'dark blue',
                                             'dark cyan',
                                             'dark green',
                                             'dark olive',
                                             'dark orange',
                                             'dark pink',
                                             'dark purple',
                                             'dark red',
                                             'dark yellow',
                                             'red and yellow stripe',
                                             'red and green stripe',
                                             'red and blue stripe',
                                             'red and purple stripe',
                                             'yellow and green stripe',
                                             'yellow and blue stripe',
                                             'yellow and purple stripe',
                                             'green and blue stripe',
                                             'green and purple stripe',
                                             'blue and purple stripe',
                                             'dark red and yellow stripe',
                                             'dark red and green stripe',
                                             'dark red and blue stripe',
                                             'dark red and purple stripe',
                                             'dark yellow and green stripe',
                                             'dark yellow and blue stripe',
                                             'dark yellow and purple stripe',
                                             'dark green and blue stripe',
                                             'dark green and purple stripe',
                                             'dark blue and purple stripe',
                                             'red and yellow polka dot',
                                             'red and green polka dot',
```

```
                                              'red and blue polka dot',
                                              'red and purple polka dot',
                                              'yellow and green polka dot',
                                              'yellow and blue polka dot',
                                              'yellow and purple polka dot',
                                              'green and blue polka dot',
                                              'green and purple polka dot',
                                              'blue and purple polka dot',
                                              'dark red and yellow polka dot',
                                              'dark red and green polka dot',
                                              'dark red and blue polka dot',
                                              'dark red and purple polka dot',
                                              'dark yellow and green polka dot',
                                              'dark yellow and blue polka dot',
                                              'dark yellow and purple polka dot',
                                              'dark green and blue polka dot',
                                              'dark green and purple polka dot',
                                              'dark blue and purple polka dot',
                                              'red swirl',
                                              'yellow swirl',
                                              'green swirl',
                                              'blue swirl',
                                              'purple swirl',
                                              'dark red swirl',
                                              'dark yellow swirl',
                                              'dark green swirl',
                                              'dark blue swirl',
                                              'dark purple swirl',
                                              'red paisley',
                                              'yellow paisley',
                                              'green paisley',
                                              'blue paisley',
                                              'purple paisley']}
```

Task 5: "Bring me a consonant that has any warm color on it"

```
In [ ]:  task_kwargs = {
             'possible_dragged_obj': ['letter G',
                                      'letter M',
                                      'letter R',
                                      'letter T',
                                      'letter V'],
             'possible_dragged_obj_texture': ['polka dot',
                                      'orange',
                                      'pink',
                                      'red',
                                      'yellow',
                                      'dark orange',
                                      'dark pink',
                                      'dark red',
                                      'dark yellow',
                                      'red and yellow stripe',
                                      'dark red and yellow stripe',
                                      'dark red and green stripe',
                                      'dark red and blue stripe',
                                      'dark yellow and purple stripe',
                                      'red and yellow polka dot',
                                      'dark red and yellow polka dot',
                                      'red swirl',
                                      'yellow swirl',
                                      'dark red swirl',
                                      'dark yellow swirl',
```

```
                                              'red paisley',
                                              'yellow paisley']}
```

## Task 6: "Bring me a vowel that has multiple colors on it" (white is not a color)

```python
In [ ]:  task_kwargs = {
             'possible_dragged_obj': ['letter A',
                                      'letter E'],
             'possible_dragged_obj_texture': ['polka dot',
                                              'checkerboard',
                                              'tiger',
                                              'magma',
                                              'rainbow',
                                              'yellow',
                                              'dark green',
                                              'dark yellow',
                                              'red and yellow stripe',
                                              'red and green stripe',
                                              'red and blue stripe',
                                              'red and purple stripe',
                                              'yellow and green stripe',
                                              'yellow and blue stripe',
                                              'yellow and purple stripe',
                                              'green and blue stripe',
                                              'green and purple stripe',
                                              'blue and purple stripe',
                                              'dark red and yellow stripe',
                                              'dark red and green stripe',
                                              'dark red and blue stripe',
                                              'dark red and purple stripe',
                                              'dark yellow and green stripe',
                                              'dark yellow and blue stripe',
                                              'dark yellow and purple stripe',
                                              'dark green and blue stripe',
                                              'dark green and purple stripe',
                                              'dark blue and purple stripe',
                                              'red and yellow polka dot',
                                              'red and green polka dot',
                                              'red and blue polka dot',
                                              'red and purple polka dot',
                                              'yellow and green polka dot',
                                              'yellow and blue polka dot',
                                              'yellow and purple polka dot',
                                              'green and blue polka dot',
                                              'green and purple polka dot',
                                              'blue and purple polka dot',
                                              'dark red and yellow polka dot',
                                              'dark red and green polka dot',
                                              'dark red and blue polka dot',
                                              'dark red and purple polka dot',
                                              'dark yellow and green polka dot',
                                              'dark yellow and blue polka dot',
                                              'dark yellow and purple polka dot',
                                              'dark green and blue polka dot',
                                              'dark green and purple polka dot',
                                              'dark blue and purple polka dot',
                                              'yellow swirl',
                                              'green swirl',
                                              'blue swirl',
                                              'dark yellow swirl',
                                              'dark green swirl',
                                              'dark blue swirl',
```

```
                                    'dark purple swirl',
                                    'red paisley',
                                    'green paisley',
                                    'blue paisley',
                                    'purple paisley']}
```

Task 7: "Bring me a letter. If there are multiple, bring the one that comes earliest in the alphabet."

```
In [ ]:  task_kwargs = {
             'possible_dragged_obj': ['letter A',
                                      'letter E',
                                      'letter G',
                                      'letter M',
                                      'letter R',
                                      'letter T'],
             'possible_dragged_obj_texture': ['brick',
                                              'tiles',
                                              'wooden',
                                              'granite',
                                              'plastic',
                                              'polka dot',
                                              'checkerboard',
                                              'tiger',
                                              'magma',
                                              'rainbow',
                                              'blue',
                                              'cyan',
                                              'green',
                                              'olive',
                                              'orange',
                                              'pink',
                                              'purple',
                                              'red',
                                              'yellow',
                                              'dark blue',
                                              'dark cyan',
                                              'dark green',
                                              'dark olive',
                                              'dark orange',
                                              'dark pink',
                                              'dark purple',
                                              'dark red',
                                              'dark yellow',
                                              'red and yellow stripe',
                                              'red and green stripe',
                                              'red and blue stripe',
                                              'red and purple stripe',
                                              'yellow and green stripe',
                                              'yellow and blue stripe',
                                              'yellow and purple stripe',
                                              'green and blue stripe',
                                              'green and purple stripe',
                                              'blue and purple stripe',
                                              'dark red and yellow stripe',
                                              'dark red and green stripe',
                                              'dark red and blue stripe',
                                              'dark red and purple stripe',
                                              'dark yellow and green stripe',
                                              'dark yellow and blue stripe',
                                              'dark yellow and purple stripe',
                                              'dark green and blue stripe',
                                              'dark green and purple stripe',
```

```
                                    'dark blue and purple stripe',
                                    'red and yellow polka dot',
                                    'red and green polka dot',
                                    'red and blue polka dot',
                                    'red and purple polka dot',
                                    'yellow and green polka dot',
                                    'yellow and blue polka dot',
                                    'yellow and purple polka dot',
                                    'green and blue polka dot',
                                    'green and purple polka dot',
                                    'blue and purple polka dot',
                                    'dark red and yellow polka dot',
                                    'dark red and green polka dot',
                                    'dark red and blue polka dot',
                                    'dark red and purple polka dot',
                                    'dark yellow and green polka dot',
                                    'dark yellow and blue polka dot',
                                    'dark yellow and purple polka dot',
                                    'dark green and blue polka dot',
                                    'dark green and purple polka dot',
                                    'dark blue and purple polka dot',
                                    'red swirl',
                                    'yellow swirl',
                                    'green swirl',
                                    'blue swirl',
                                    'purple swirl',
                                    'dark red swirl',
                                    'dark yellow swirl',
                                    'dark green swirl',
                                    'dark blue swirl',
                                    'dark purple swirl',
                                    'red paisley',
                                    'yellow paisley',
                                    'green paisley',
                                    'blue paisley',
                                    'purple paisley']}
```

Task 8: "Bring me something I can drink water out of"

```
In [ ]:  task_kwargs = {
             'possible_dragged_obj': ['bowl',
                                      'container'],
             'possible_dragged_obj_texture': ['brick',
                                              'tiles',
                                              'wooden',
                                              'granite',
                                              'plastic',
                                              'polka dot',
                                              'checkerboard',
                                              'tiger',
                                              'magma',
                                              'rainbow',
                                              'blue',
                                              'cyan',
                                              'green',
                                              'olive',
                                              'orange',
                                              'pink',
                                              'purple',
                                              'red',
                                              'yellow',
                                              'dark blue',
                                              'dark cyan',
```

```
'dark green',
'dark olive',
'dark orange',
'dark pink',
'dark purple',
'dark red',
'dark yellow',
'red and yellow stripe',
'red and green stripe',
'red and blue stripe',
'red and purple stripe',
'yellow and green stripe',
'yellow and blue stripe',
'yellow and purple stripe',
'green and blue stripe',
'green and purple stripe',
'blue and purple stripe',
'dark red and yellow stripe',
'dark red and green stripe',
'dark red and blue stripe',
'dark red and purple stripe',
'dark yellow and green stripe',
'dark yellow and blue stripe',
'dark yellow and purple stripe',
'dark green and blue stripe',
'dark green and purple stripe',
'dark blue and purple stripe',
'red and yellow polka dot',
'red and green polka dot',
'red and blue polka dot',
'red and purple polka dot',
'yellow and green polka dot',
'yellow and blue polka dot',
'yellow and purple polka dot',
'green and blue polka dot',
'green and purple polka dot',
'blue and purple polka dot',
'dark red and yellow polka dot',
'dark red and green polka dot',
'dark red and blue polka dot',
'dark red and purple polka dot',
'dark yellow and green polka dot',
'dark yellow and blue polka dot',
'dark yellow and purple polka dot',
'dark green and blue polka dot',
'dark green and purple polka dot',
'dark blue and purple polka dot',
'red swirl',
'yellow swirl',
'green swirl',
'blue swirl',
'purple swirl',
'dark red swirl',
'dark yellow swirl',
'dark green swirl',
'dark blue swirl',
'dark purple swirl',
'red paisley',
'yellow paisley',
'green paisley',
'blue paisley',
'purple paisley']}
```

# Task 9: "Bring me something I can find in a typical kitchen"

```python
task_kwargs = {
    'possible_dragged_obj': ['bowl',
                             'container',
                             'pan'],
    'possible_dragged_obj_texture': ['brick',
                                      'tiles',
                                      'wooden',
                                      'granite',
                                      'plastic',
                                      'polka dot',
                                      'checkerboard',
                                      'tiger',
                                      'magma',
                                      'rainbow',
                                      'blue',
                                      'cyan',
                                      'green',
                                      'olive',
                                      'orange',
                                      'pink',
                                      'purple',
                                      'red',
                                      'yellow',
                                      'dark blue',
                                      'dark cyan',
                                      'dark green',
                                      'dark olive',
                                      'dark orange',
                                      'dark pink',
                                      'dark purple',
                                      'dark red',
                                      'dark yellow',
                                      'red and yellow stripe',
                                      'red and green stripe',
                                      'red and blue stripe',
                                      'red and purple stripe',
                                      'yellow and green stripe',
                                      'yellow and blue stripe',
                                      'yellow and purple stripe',
                                      'green and blue stripe',
                                      'green and purple stripe',
                                      'blue and purple stripe',
                                      'dark red and yellow stripe',
                                      'dark red and green stripe',
                                      'dark red and blue stripe',
                                      'dark red and purple stripe',
                                      'dark yellow and green stripe',
                                      'dark yellow and blue stripe',
                                      'dark yellow and purple stripe',
                                      'dark green and blue stripe',
                                      'dark green and purple stripe',
                                      'dark blue and purple stripe',
                                      'red and yellow polka dot',
                                      'red and green polka dot',
                                      'red and blue polka dot',
                                      'red and purple polka dot',
                                      'yellow and green polka dot',
                                      'yellow and blue polka dot',
                                      'yellow and purple polka dot',
                                      'green and blue polka dot',
                                      'green and purple polka dot',
```

```
                                          'blue and purple polka dot',
                                          'dark red and yellow polka dot',
                                          'dark red and green polka dot',
                                          'dark red and blue polka dot',
                                          'dark red and purple polka dot',
                                          'dark yellow and green polka dot',
                                          'dark yellow and blue polka dot',
                                          'dark yellow and purple polka dot',
                                          'dark green and blue polka dot',
                                          'dark green and purple polka dot',
                                          'dark blue and purple polka dot',
                                          'red swirl',
                                          'yellow swirl',
                                          'green swirl',
                                          'blue swirl',
                                          'purple swirl',
                                          'dark red swirl',
                                          'dark yellow swirl',
                                          'dark green swirl',
                                          'dark blue swirl',
                                          'dark purple swirl',
                                          'red paisley',
                                          'yellow paisley',
                                          'green paisley',
                                          'blue paisley',
                                          'purple paisley']}
```