# OpenReview forum: "Learning with Language-Guided State Abstractions"
_ICLR.cc/2024/Conference — ICLR 2024 poster_

### Official Review · Reviewer_4L8o · 2023-10-25

**Soundness:** 3 good
**Presentation:** 3 good
**Contribution:** 2 fair
**Rating:** 5
**Confidence:** 5

**Summary:**

This paper introduces an LLM-based approach to mask out task-irrelevant objects in visual observations to improve imitation learning. The resulting “abstract states” - observations with only task-relevant features, enables training behavior cloning policies with better data efficiency and generalization. The method is straightforward: taking a task specification and the observed object information as input, a pretrained LLM is leveraged to remove irrelevant objects. Optionally, the authors recruited human subjects to refine the LLM output or identify irrelevant objects from scratch as ablations.
The authors evaluate the proposed approach on the VIMA[1] benchmark, comparing it with behavior cloning policies trained with various forms of state abstractions. The results indicate that the proposed method: 1) enhances data efficiency and success rate while requiring less human efforts; and 2) can generalize to unseen object textures, distractor objects, and task specifications.

[1] Jiang, Yunfan, et al. "Vima: General robot manipulation with multimodal prompts." ICML 2023.

**Strengths:**

1. The idea to leverage the semantic reasoning ability of LLMs to identify task-relevant features is interesting and innovative. The approach of masking out irrelevant objects does prevent learning robot policies harmed by spurious correlations.
2. The paper is written clearly and easy to follow.
3. The paper includes comprehensive experiments to demonstrate the effectiveness of the proposed method.

**Weaknesses:**

1. The primary weakness of the paper lies in the absence of a lack of substantial technical contribution. While the proposed strategy mitigates the issue of spurious correlations, it appears more as a choice of system design to generate heuristics rather than a fundamental method to identify useful state features. Notably, the importance of state features not only depends on task semantics, but also the low-level geometric constraints imposed by the environment and robot embodiment. For example, some objects not mentioned in the task specification may be important for robot collision avoidance. Consequently, the learning of important state features and motion policies are coupled and should ideally be learned together, such as in VIOLA [2].
2. Another drawback is that the imitation learning setup is overly simplified . The use of VIMA [1] benchmark, which employs high-level primitive actions like “pick” and “place” with continuous goal poses, significantly simplifies the training of a behavior cloning policy. Given that the major objective is to assess the proposed state abstraction in imitation learning, it would be advisable for the authors to consider a more rigorous robot manipulation benchmark with continuous actions, such as RoboSuite [3].
3. State abstraction is a fundamental problem in decision making. I think it would be helpful for the readers to understand the context better if the authors refer and discuss the related literature (such as [4,5]).


[2] Zhu, Yifeng, et al. "Viola: Imitation learning for vision-based manipulation with object proposal priors." CoRL 2022.
[3] https://robosuite.ai/
[4] Li, Lihong, Thomas J. Walsh, and Michael L. Littman. "Towards a unified theory of state abstraction for MDPs." AI&M 2006.
[5] Tomar, Manan, et al. "Model-invariant state abstractions for model-based reinforcement learning." ICLR 2022.

**Questions:**

Regarding the observation input to behavior cloning in LGA, do you use the binary mask directly (as visualized in Figure 1) or the masked image? If you use the former one, I wonder how the latter one works.

---

> ### Author Response · Authors · 2023-11-16
> **Author Response**
>
> We thank the reviewer for their feedback!
>
> # Significance of method
>
> > it appears more as a choice of system design to generate heuristics rather than a fundamental method
>
> We respectfully disagree with the reviewer that, because we are demonstrating a capability of applying language priors to imitation learning, our contribution is not “a fundamental method”.
>
> First, as we know from the recent boom of LM+robotics papers, the premise of applying strong pretrained language priors to problems of decision-making is of high value and immensely impactful to the field. We simply take an alternative approach from those of previous methods: rather than leveraging LMs for *action abstraction* i.e. action stitching as in SayCan, we use them for *state abstraction*, i.e. identifying relevant features. LGA therefore demonstrates the utility of strong LM priors in a novel facet of decision-making.
>
> Second, we agree that VIOLA is an important related work and will update our paper to discuss it in more detail. However, it is not evident to us that learning important state features should *ideally* be learned with the motion policy: indeed, this forces the agent to re-learn motor policies for novel tasks (e.g. a new linguistic specification), whereas our approach allows the re-use of motor policies across tasks (if the new linguistic utterance has been found to be the same as previously learned task with an alternative specification). However, we believe that learning how to blend these different state representations (both abstract and raw) is an important direction for future work, and will emphasize in the discussion.
>
> Finally, we evaluate LGA with BC here since it is the clearest, simplest method for isolating the impact of good abstractions, but it is entirely feasible to utilize the abstraction generation process of LGA in tandem with a method like VIOLA for jointly learning well-attended policies (e.g. by also conditioning the transformer-based policy in VIOLA on LGA-generated abstractions). We are happy to clarify this scoping of LGA in the new revision.
>
> # Evaluation
>
> > VIMA is too simple
>
> We appreciate the reviewer for pointing out the limitations of VIMA as an evaluation domain, and for the suggestion of RoboSuite. Although evaluating on harder benchmarks is a great suggestion for future work, we felt it prudent to instead spend time on demonstrating the practical feasibility of LGA on a real robot. As we wrote about in Sec. 7, we demonstrated the full LGA pipeline on a Spot robot, including using raw robot observations segmented via SAM, captioned via Dedic, and then imitation learned on real user tele-op’d trajectories. Consistently (~9/10 test tasks), the state abstractions produced via LGA were able to produce policies successful at completing the tasks. The few failure cases observed were due to the captioning model failing (e.g. SAM reliably detected different shapes and sizes of water bottles but Dedic was unable to always produce the text description “water bottle”).
>
> Thus, the LGA pipeline can be reliably deployed in real-world scenarios, and we expect that it can be straightforwardly generalized to even more complex ones, especially as individual components of the pipeline improve. While we can always evaluate on harder/alternative simulated benchmarks, we hope the reviewer appreciates the effort spent on ensuring the full pipeline is able to be practically implemented with real hardware!
>
> # Question
>
> > do you use the binary mask directly (as visualized in Figure 1) or the masked image?
>
> We used the binary mask directly, although the reviewer suggests a reasonable alternative. In our approach, we explicitly highlighted the flexibility of LGA to operate on both the binary mask as well as one conditioned on the raw observation, although note we did not include masked image as an alternative. We suspect that for tasks where texture/color is not meaningful, e.g. once the system has detected it’s an image, there’s no need to preserve the color “orange” in the observation”, although either would likely work. We’re happy to clarify this additional option in the revision.
>
> # Missing Literature
>
> We thank the reviewer for bringing missing references to our attention, for they indeed help frame the historical context of state abstraction in decision-making! We will add them to the related work.

---

> > ### Comment · Reviewer_4L8o · 2023-11-22
> >
> > I would like to thank the authors for their detailed explanations and replies.
> >
> > I agree that the major contribution of this paper is to demonstrate the new capability that language priors can facilitate imitation learning. However, I still think the overly-simplified experimental setup with primitive actions and binary mask states is insufficient to support the claim, especially as most of the VIMA tasks can be solved with a sequence of 1-2 primitive actions. Though the authors learn teleoperated trajectories in real-robot experiments, the quantitative results are not reported.
> >
> > I'm happy to raise my rating to 5. In general I like this paper for its clear motivation, clean method and comprehensive discussions. It will be great if more work can be done to evaluate the method in more rigorous imitation learning setups.

---

> ### Author Response · Authors · 2023-11-21
> **Followup**
>
> Dear Reviewer 4L8o,
>
> Thank you for your review and feedback!
>
> As the discussion period is coming to a close, please let us know if our response has adequately addressed your concerns, or if you have any remaining questions. If not, we would appreciate if you could raise your score. Thanks for your work in serving as a reviewer!
>
> Thanks,
> Authors

---

### Official Review · Reviewer_PEPS · 2023-10-30

**Soundness:** 2 fair
**Presentation:** 3 good
**Contribution:** 2 fair
**Rating:** 6
**Confidence:** 3

**Summary:**

This work presents a method LGA that leverages language models to compose abstract states for few-shot imitation learning. Visual RGB observations are first segmented and textualized into features that a language model is then tasked to filter conditioning on the language instruction. The policy is then trained with such abstract visual state.

Experiments in a simulated benchmark shows the proposed method outperforms naive baselines.

**Strengths:**

This work presents an intuitive and simple idea that is shown to be powerful for few-shot imitation learning of policies that are specifically designed to generalize across variations of color and texture.

The proposed method leverages the commonsense reasoning capabilities of large language models for reducing the task complexity for imitation learning, which is an exciting application of pretrained language models in robotics.

**Weaknesses:**

Conceptually, the idea of using visual masks as attention or part of state representation isn’t novel and has been explored in various prior works including recent ones such as robotmoo[1] and VIOLA[2].

Feature abstraction of LGA takes a filtering approach that relies on segmentation and textualization operates at the desired abstraction level and is complete. For example, language instruction can be about a group of objects, the object as a whole or only part of an object and it is unclear how to segment or group segmentations before we know the task.
An alternative approach to filtering would be leveraging open-vocabulary object detectors or VLMs for identifying the target objects like in recent works using LLMs for planning, which the paper didn’t ablate.

Using binary masks as state representation seems to be limiting and can hurt in tasks where the texture or details of the object matters, maybe the language model should decide if the binary mask or original state should be used or not based on the context. On a similar note, it seems from the results LGA-S performs better anyway?

At the same time the background might be important landmarks for the robot to understand relative size. For example if the robot is learning to visually navigate to certain objects, this proposed method would fail if the model removes all the background necessary for the robot to localize itself.

The authors should explicitly discuss assumptions and limitations of the method to specific types of tasks/settings.

[1] https://robot-moo.github.io/
[2] https://ut-austin-rpl.github.io/VIOLA/



---Edit----

The rebuttal addressed some of my initial concerns and I appreciate the response and explanation from the authors. I do think demonstrating the failure modes of LGA and discussing the limitations is as important as showing the positive results.

I am happy to raise my evaluation but won't argue strongly for acceptance.

**Questions:**

It seems LGA relies heavily on the segmentation and captioning module. How well does these systems work? What are some common failure modes? Can the robot arm be successfully segmented?

Does LGA or the instantiation assume full observability? or does it run segmentation and feature extraction on each and every frame? this seems expensive given the size of SAM

How good is the language model at guessing relevant objects if they are not explicitly mentioned in the language instruction?

---

> ### Author Response · Authors · 2023-11-16
> **Author Response**
>
> We thank the reviewer for their feedback!
>
> # Questions
>
> > Does LGA assume full observability or run segmentation on every frame?
>
> In Sec. 7, we explored the feasibility of implementing the full LGA pipeline on a real robotic system: a Spot robot with arm, real segmentation via SAM, and real captioning with Dedic. In order to avoid the real-time latency of running SAM+LM on every timeframe (as the reviewer suggests), we did assume full observability of the scene at the initial timestep, which is a common assumption many vision-based robotics works also make (e.g. the robot needs to see objects in order to manipulate them). However, if/when running inference of such models becomes less expensive, it’s certainly feasible to apply LGA at every timestep, or every *k* timesteps as needed!
>
> > How well does [segmentation and captioning] work in the real world? What are common failure modes?
>
> As noted above, real-world segmentation + captioning worked quite well on the real robot, reliably able to create good state abstractions successfully ~9/10 times for each scenario (bringing different kinds of fruit and throwing away different types of cans/bottles). The few failure cases observed were due to the captioning model failing (e.g. SAM reliably detected different shapes and sizes of water bottles but Dedic was unable to always produce the text description “water bottle”). As a result, we are quite hopeful that the LGA pipeline can be extended to additional real-world scenarios, especially as individual components of the pipeline improve.
>
> > How good is the LM at guessing relevant objects not explicitly mentioned in the language instruction?
>
> The reviewer is correct that potential task-relevant features must be referenced in the user’s language utterance (e.g. “pick up the fruit while avoiding obstacles” if object-avoidance is sometimes important to the task, sometimes not). However, LGA does afford a degree of flexibility not captured by prior methods, i.e. LGA does not require objects to be mentioned *by name*. While we did not test this directly, we did observe instances of this: for example, LGA was able to convert “something I can put food in'' to the object "bowl" and "ripe fruit” to “yellow banana” but not “green banana”. We will add these to the appendix and update the discussion to include this as an exciting future direction!
>
> # Significance of approach
>
> > Conceptually, the idea of using visual masks as attention or part of state representation isn’t novel and has been explored in various prior works including recent ones such as robotmoo[1] and VIOLA[2]
>
> We agree that the idea of using visual masks as state representation is not new, and did not intend to claim it as a contribution. LGA demonstrates the feasibility of leveraging *language priors* to *create* these state abstractions. This is distinct from VIOLA, which explores the feasibility of leveraging *image priors*.
>
> While MOO takes a step closer to our approach by leveraging vision-language priors, we note that first, the paper was arxiv’d a month after the ICLR deadline (and therefore is concurrent work to ours) and second, that their approach relies on models that perform caption-matching from datasets already grounded in real-world objects (as opposed to ours, which leverages language-only priors to flexibly map ANY language-specified feature into the scene, not just the goal object).
>
> > language instruction can be about a group of objects, the object as a whole or only part of an object and it is unclear how to segment or group segmentations before we know the task.
>
> We certainly agree that these are important scenarios! Segmentation should ideally be fine-grained, as the LM can only be queried about segmented and captioned features. However, given fine-grained segmentation on the level of object(s), LGA can certainly handle situations such as “a group of objects”. For example, consider a task like “bring me all fruits from the table”, where both a tomato and a banana are present in the scene. We verified through a quick LM experiment that the abstraction created by the LGA would include both the tomato and the banana since the LM is queried about whether each individual feature in the scene is relevant to the specified task! We believe it’s this flexibility that differentiates LGA from VLM or alternative attention-based approaches for imitation learning, since the LM can flexibly construct ANY abstraction from text, not just for specific skills like “pick up [x] object”.

---

> > ### Author Response · Authors · 2023-11-16
> > **Author Response Pt 2**
> >
> > # Including the state
> >
> > > maybe the language model should decide if the binary mask or original state should be used or not based on the context
> >
> > We agree with the reviewer that this is an important decision. We believe the question of “*when* to use abstractions vs. not” is orthogonal to our work, which explores “*how* to use language to construct good state abstractions.”  Thus, in our work, we explicitly highlighted the flexibility of LGA to operate on both the binary mask as well as the raw observation. We believe exploring this tradeoff further is an important direction for future work – and particularly appreciate the reviewer’s suggestion to allow the LM to decide! We will update the discussion to reflect this.

---

### Official Review · Reviewer_rEzX · 2023-11-02

**Soundness:** 3 good
**Presentation:** 3 good
**Contribution:** 3 good
**Rating:** 6
**Confidence:** 4

**Summary:**

This paper introduces the Language-Guided Abstraction (LGA) framework, which utilizes natural language to construct state abstraction for imitation learning. The method comprises three key steps: First, in the textualization phase, it transforms raw perceptual input into a text-based feature set. Second, during the state abstraction step, a pre-trained language model is employed to filter out irrelevant features from the feature set, creating task-specific state abstractions. Finally, in the instantiation stage, the reduced abstracted feature set is converted into a format understandable by the policy, such as an observation displaying only the pertinent objects.

**Strengths:**

LGA avoids spurious correlations by highlighting goal information in semantic maps, not raw pixels. LGA converts language and observations into unambiguous states to enhance policy adaptability. This is especially important when only limited training data is available. The integration with Language Models enables contextually appropriate task-relevant feature selection, boosting the overall policy generalization and performance at test time.

The experiment results demonstrate that LGA reduces the time needed for feature specification compared to manual methods, yet outperforms non-abstraction-based baselines in terms of sample efficiency. Policies trained using LGA's state abstractions exhibit resilience to observational shifts and language variations. In multi-task scenarios, LGA effectively resolves task ambiguities and adapts to new language specifications in observations.

**Weaknesses:**

There appears to be a gap in the evaluation regarding task failures—whether they stem from policy quality or incorrect state abstraction remains unclear. The experiment results do not specify how frequently the language model predicts insufficient or redundant state abstraction, and whether refining its choices with feedback from policy execution is a feasible solution remains unexplored.

The paper appears to overlook extensive research on learning state abstraction for reinforcement learning, including notable works such as "Approximate State Abstraction" (ICML 2016), "State Abstraction In Lifelong RL" (ICML 2018), and "State Abstraction As Compression" (AAAI 2019).

**Questions:**

Can you categorize task failures into two groups: those caused by policy quality and those resulting from incorrect state abstraction?

---

> ### Author Response · Authors · 2023-11-16
> **Author Response**
>
> We thank the reviewer for their feedback!
>
> # Reviewer Question
>
> > Can you categorize task failures into two groups: those caused by policy quality and those resulting from incorrect state abstraction?
>
> The experiments we ran were intended to isolate the effect of good abstractions generated via LGA vs. bad abstractions (i.e. raw observations containing spurious features). Consider the tasks shown in Fig. 4. For each task, we fixed the training and test distribution for demonstrations for all LGA variants and the GCBC baseline. In other words, we held the training data, test data, and model training constant, varying only whether the model was trained on the LM-generated abstraction vs. the raw observation (which intuitively is a bad “abstraction” containing many spurious features). The resulting performance gap can therefore be attributed to the quality of the abstractions used for learning. Our hypothesis is that bad abstractions (containing irrelevant features) will perform poorly when confronted with covariate shifts such as new colors or distractors. We note this evaluation is similar to work in reward learning, which often assesses the quality of the learned reward via policy success on the downstream task.
>
> That said, we agree with the reviewer’s point that the current results do not explicitly specify the correctness of abstraction generation. To verify that our evaluation above does indeed capture the effect we claim, we have defined an additional metric assessing similarity of the LM generated and  human-designed feature sets.  We measure the Hamming distance, where smaller values indicate similarity. As an example, for the four tasks in Fig. 4A, the following results show that 75% of the time, LGA generated feature sets match those of human engineered sets when compared to those from raw observations.
>
> | Task  | LGA | Observation |
> | ----------- | ----------- | ----------- |
> | heart (pp) | 0 | 24 |
> | letter (pp) | 0 | 22 |
> | bowl (rotate) | 0 | 22 |
> | block (sweep, avoid) | 6 | 48 |
>
>
>
> # Iterative Abstraction Construction
>
> > whether refining its choices with feedback from policy execution is a feasible solution remains unexplored.
>
> We think this is an excellent suggestion, and one that we are currently exploring in future work! Notably, for this type of iterative abstraction update to be feasible, one must assume a way of generating rewards/scores for judging policy rollouts. We did not feel iterative updates were central to the idea of LGA (which explored the feasibility of leveraging LM priors for constructing state abstractions, not for scoring policy rollouts), so we decided to separate out into future work. We are happy to discuss this choice in the discussion!
>
> # Missing literature
>
> > The paper appears to overlook extensive research on learning state abstraction for reinforcement learning
>
> Thank you for highlighting these relevant and important works. We note that 2 were already cited in the paper (see the two Abel et al. papers on p. 1 and 3). We agree the Abel et al, 2019 paper is also relevant: briefly, LGA differs because it constructs such abstractions via *language-guided priors*. We will update the paper to incorporate and discuss this work.

---

> ### Comment · Reviewer_rEzX · 2023-11-21
> **Thanks for the authors' response**
>
> I remain positive after reading the rebuttal. The paper introduces a simple approach to leverage LLM for effective robot learning by state abstraction. This method sets it apart from existing approaches, which primarily use LLM for planning and reward learning.

---

### Meta-Review · Area_Chair_LMyq · 2023-12-06

**Metareview:**

(a) Scientific Claims and Findings:
The paper introduces the Language-Guided Abstraction (LGA) framework for imitation learning. LGA employs natural language processing to construct task-specific state abstractions from raw perceptual input, focusing on filtering out irrelevant features. The method involves textualizing perceptual input into text-based features, employing a pre-trained language model for feature selection, and imitating using these learned features. The paper claims that LGA enhances policy adaptability, especially with limited training data, by using language models for contextually appropriate feature selection. Experimentally, LGA reportedly demonstrates resilience to observational shifts and language variations and outperforms non-abstraction baselines in sample efficiency.

(b) Strengths:
(+) Novel Application of Language Models (Reviewer rEzX, PEPS): LGA's use of language models for state abstraction is conceptually new, differentiating it from existing approaches that primarily use LLMs for planning and reward learning.

(+) Effective Handling of Sparse Data and Generalization (Reviewer rEzX): The framework is shown to handle sparse training data and shows potential in generalizing across task variations.

(+) Clarity and Presentation (Reviewer 4L8o): The paper is clear and easy-to-follow.

(c) Weaknesses:
(-) Technical Depth and Novelty (Reviewer 4L8o, rEzX, PEPS): The paper has limited technical contribution, with feature masking not being a novel concept. In-depth analysis of where the approach is expected to work and fail is missing.

(-) Simplicity of Experimental Setup (Reviewer 4L8o): The use of the VIMA benchmark and the absence of more complex, real-world testing scenarios is a limitation. More complex robot manipulation benchmarks are needed.

(-) Omission of Relevant Literature (Reviewer rEzX): There's a noted gap in acknowledging significant prior work in learning state abstraction for reinforcement learning.

(-) Unclear Evaluation Metrics (Reviewer rEzX): The paper does not adequately address how often the language model predicts insufficient or redundant state abstraction, nor does it discuss refining choices with feedback from policy execution.

**Justification For Why Not Higher Score:**

The paper does not have very strong technical contributions and the evaluations are simple.

**Justification For Why Not Lower Score:**

The reviewers feel that the paper is well written, the idea is still conceptually novel and the merits outweigh the weakness

---

### Decision · Program_Chairs · 2024-01-16

Accept (poster)